

# Deep learning-based method for sentiment analysis for patients' drug reviews

Sena Al-Hadhrami[1], Tamas Vinko[1], Tawfik Al-Hadhrami[2], Faisal Saeed[3] and Sultan Noman Qasem[4]

[1] Institute of Informatics, Faculty of Science and Informatics, University of Szeged, Szeged, Hungary
[2] Computer Science Department, School of Science and Technology, Nottingham Trent University, Nottingham, United Kingdom
[3] DAAI Researsh Group, College of Computing and Digital Technology, Birmingham City University, Birmingham, United Kingdom
[4] Computer Science Department, College of Computer and Information Sciences, Imam Mohammad Ibn Saud Islamic University (IMSIU), Riyadh, Saudi Arabia

Corresponding authors
Sena Al-Hadhrami,
senaalhadhrami@gmail.com
Tawfik Al-Hadhrami,
tawfik.al-hadhrami@ntu.ac.uk

## ABSTRACT

This article explores the application of deep learning techniques for sentiment analysis of patients' drug reviews. The main focus is to evaluate the effectiveness of bidirectional long-short-term memory (LSTM) and a hybrid model (bidirectional LSTM-CNN) for sentiment classification based on the entire review text, medical conditions, and rating scores. This study also investigates the impact of using GloVe word embeddings on the model's performance. Two different drug review datasets were used to train and test the models. The proposed methodology involves the implementation and evaluation of both deep learning models with the GloVe word embeddings for sentiment analysis of drug reviews. The experimental results indicate that Model A (Bi-LSTM-CNN) achieved an accuracy of 96% and Model B (Bi-LSTM-CNN) performs consistently at 87% for accuracy. Notably, the incorporation of GloVe word representations improves the overall performance of the models, as supported by Cohen's Kappa coefficient, indicating a high level of agreement. These findings showed the efficacy of deep learning-based approaches, particularly bidirectional LSTM and bidirectional LSTM-CNN, for sentiment analysis of patients' drug reviews.

## INTRODUCTION

Sentiment analysis, also known as opinion mining (OM), has gained significant popularity as a crucial technique in natural language processing. Its objective is to extract subjective information from text data, encompassing opinions, emotions, and attitudes (*Pang, Lee & Vaithyanathan, 2002*). With the rise of online healthcare platforms, there has been an exponential increase in patients' drug reviews. In response, medical institutions have established dedicated offices to manage patient experiences, ensuring that patients are informed about the potential risks and complications associated with treatments. Hence,

sentiment analysis of patients' drug reviews holds immense value in providing insights into patient satisfaction, drug efficacy, and adverse drug reactions. This information assists decision-makers in the healthcare sector to gauge the quality of services and address potential patient issues. Two primary approaches to sentiment analysis have been explored in *Aung & Myo (2017)*. The first is lexicon-based, which relies on dictionaries or lexicons containing words and their corresponding sentiments. The second approach involves machine learning algorithms that learn sentiments from labeled datasets. While several attempts have been made using machine learning techniques, only a few studies have applied deep learning to sentiment analysis of drug reviews. This work aims to fill that gap by investigating and evaluating various deep learning architectures, such as convolutional neural network (CNN), and long-short term memory (LSTM), with a focus on word embeddings using Global Vectors for Word Representation (GloVe), which represents words as high-dimensional vectors.

The sentiment analysis process involves several steps, including data cleaning and the addition of extra-linguistic constraints, such as negation handling, on the pre-trained embeddings. Deep learning techniques, including recurrent neural networks (RNNs), LSTM, and encoder representations from transformers models, are employed for the sentiment analysis task. Standard evaluation metrics like accuracy, precision, recall, and F1 are utilized to assess the performance of the systems. Contextual understanding means that the meaning of a word can change depending on the context in which it is used. So, understanding the context in which the word is used is essential to accurately identify the sentiment. *Zhao & Mao (2017)* demonstrated the limitations of the bag-of-words text encoding without considering the semantics/context of different words in a text document. These limitations proved to be a major setback for the natural language processing (NLP) tasks which required diverse vocabulary and contextual encoding. To overcome these limitations, the multi-label text classifications based on deep learning algorithms have been used in this work which was already trained on large text corpus using semantic encoding. The well-known encoding is GloVe word embedding. Therefore, this particular study has three primary objectives: Firstly, it investigates the effectiveness of the deep learning algorithms depending on the classification of sentiment analysis for drug review text, conditions, and the rating score. Secondly, we compare the effectiveness of bidirectional long short-term memory (Bi-LSTM), and hybrid model (bidirectional LSTM-CNN). We develop a deep learning-based algorithm for sentiment analysis by involving the word embedding and GloVe learning algorithms on the third objective. Lastly, we evaluate and test the performances of the developed model by using a number of metrics such as accuracy, precision, recall, and F1-score. The recent studies by *Ahmet & Abdullah (2020)* revolve around the exploration and analysis of current trends and advances in sentiment analysis, specifically focusing on deep learning-based approaches. The central theme of another study by *Zhang et al. (2022)* revolves around aspect-based sentiment analysis (ABSA), a significant fine-grained sentiment analysis problem that has garnered substantial attention over the past decade.

The report structure comprises various sections, starting with the introduction, which highlights the need for this research. The literature survey section provides a concise

overview of previous studies in this area. The methodology section describes the methods adopted for the study, followed by the results section, which evaluates the applied deep learning algorithms and word embedding approaches. Finally, the conclusion section summarizes the findings of the study.

## RELATED WORK

The research presented by *Na et al. (2012)* proposes a rule-based linguistic approach for sentiment classification of drug reviews. The clause-level sentiment classification algorithm is developed and applied to drug reviews on a discussion forum. The algorithm adopts a purely linguistic approach to computing the sentiment of a clause from the polarity with three values positive, negative, and neutral assigned to individual words, taking into consideration the grammatical dependency structure of the clause using the sentiment analysis rules. MetaMap, a medical resource tool, is used to identify various disease terms in the review documents to utilize domain knowledge for sentiment classification. Experiment results with 1,000 clauses show the effectiveness of the proposed approach, and it performed significantly better than baseline machine learning approaches. They compared their rule-based approach with a machine learning approach using naïve Bayes and found that their rule-based approach achieved comparable performance. The authors also performed an analysis of the rules to identify the most significant rules for sentiment classification. The approach is dependent on the development of a comprehensive set of rules that require domain expertise, and may not be able to capture the nuances of sentiment expressed in natural language. More recently, deep learning models have shown remarkable results for sentiment analysis on drug reviews. *Colón-Ruiz & Segura-Bedmar (2020)* studied the performance of CNN and LSTM. According to *Balakrishnan et al. (2022)*, the utilization of bidirectional encoder representations from transformers (BERT) in conjunction with Bi-LSTM was employed for conducting sentiment analysis on drug reviews. This study highlights the integration of state-of-the-art language representation models and recurrent neural networks in the domain of drug review sentiment analysis. Another investigation conducted by *Dang, Moreno-García & De la Prieta (2021)* demonstrated the effectiveness of the hybrid CNN-LSTM architecture as an approach for sentiment analysis on drug reviews. This study provides insights into the application of advanced models in sentiment analysis within the context of drug-related content.

The research by *Na & Kyaing (2015)* focused similarly on the tasks discussed earlier, there has been a proliferation of research applying deep learning models to sentiment analysis in recent years. For instance, *Abdualgalil et al. (2022)* applies several machine learning and deep learning methods to aid governments and health policymakers in making informed decisions to reduce tuberculosis prevalence through efficient forecasting methods. These methods include SARIMAX, LSTM, CNN-LSTM Hybrid, MLP network, SVR, XGboost, and RF Regression to forecast pulmonary positive, negative, and overall tuberculosis incidence cases. In addition, *Moradzadeh et al. (2021)* underscores the exceptional performance of the Bi-LSTM technique, demonstrating a superior correlation coefficient compared to alternative methods in the realm of short-term load forecasting.

The GloVe is a successful word embedding algorithm that helps to achieve high accuracy. GloVe word embedding is a global log-bilinear regression model and is based on the co-occurrence and factorization of a matrix to get vectors. *Rezaeinia, Ghodsi & Rahmani (2017)* worked on improving the accuracy of the well-known pre-trained word embeddings for sentiment analysis. They achieved a high accuracy of pre-trained word embeddings based on the combination of four approaches such as lexicon-based approach, POS tagging approach, word position algorithm, and Word2Vec/GloVe approach.

*Shilpa et al. (2021)* considered the use of GloVe word embedding with the DBSCAN clustering algorithm in document clustering. The preprocessing is done with and without stemming from Wikipedia and IMDB datasets. The GloVe word embedding algorithm was applied with the DBSCAN clustering algorithm.

*Lauren et al. (2018)* proposed the evaluating word embedding models and methods. The experimental results showed how the pertaining by using the GloVe affected the performance of the model.

In addition, *Maas et al. (2011)* presented a vector space model that learns word representations capturing semantic and sentiment information. The model's probabilistic foundation gives a theoretically justified technique for word vector induction as a factorization-based technique commonly used.

According to *Na & Kyaing (2015)*, focused on sentiment analysis of user-generated content on drug review websites. The authors used a machine learning approach, specifically a support vector machine (SVM), to classify drug reviews into positive, negative, or neutral sentiment categories. The authors also conducted feature selection to identify the most important features for sentiment classification. So, the studies also highlight the importance of feature selection and domain expertise in developing effective sentiment classification models.

*Pang, Lee & Vaithyanathan (2002)* demonstrated the effectiveness of machine learning approaches for sentiment classification and highlights the importance of feature selection and domain expertise in developing effective sentiment classification models. The study also provided a benchmark dataset for sentiment classification that has since been used in many subsequent studies.

The research done in *Mishra, Malviya & Aggarwal (2015)* used the drug reviews written by patients in different health communities to recognize commonly occurring problems. Then they compare these issues with the drug labels approved by the Food and Drug Administration (FDA) to identify any areas for improvement. They develop a system that can be scaled up for mapping interventions to indications and the associated symptoms mentioned in patient comments. By creating these mappings, our system can compare various sections of the FDA labels and provide recommendations. To give an overall rating to the drugs, it has been used an SVM-based framework was used for sentiment analysis.

Multiple aspects related to drugs have been studied in several studies that have focused on aspect-based sentiment analysis, with the aim of developing an effective sentiment classification approach. *Bobicev et al. (2012)* proposed a bag-of-words (BoW) approach to represent Twitter messages that disclose personal health information. The authors experimented with various machine learning algorithms such as naive Bayes, decision

trees, KNN, and SVM. Similarly, in *Ali et al. (2013)* several algorithms including naive Bayes, SVM, and logistic regression were investigated to estimate the polarity of patients' posts in online health forums. In the context of the study, *Wilson, Wiebe & Hoffmann (2005)* provided a foundation for the employed sentiment analysis features. These features encompassed metrics such as the number of subjective words, adjectives, adverbs, pronouns, and the presence of positive, negative, and neutral words were used to train the algorithms. These features were taken from the Subjectivity Lexicon (*Wilson, Wiebe & Hoffmann, 2005*).

*Yadav & Vishwakarma (2020)* proposed a weighted text representation framework that utilizes feature selection to reduce the dimensionality of the data and a weighting scheme that assigns weights to different words based on their importance in predicting the sentiment. The authors evaluate the performance of their framework using a dataset of drug reviews from the social media platform Twitter. They compared the performance of their framework with traditional BoW and TF-IDF methods, as well as with several machine learning algorithms including naive Bayes, support vector machines (SVM), and decision trees.

To summarise, natural language processing (NLP) has been regarded as a pivotal element of artificial intelligence (AI). This perspective stems from the fact that comprehending and producing natural language serves as a sophisticated benchmark of intelligence. As a potent instrument within the AI toolkit, deep learning, a potent instrument within the AI toolkit, also finds its place within the broader spectrum of AI endeavors. Subsequently, we delve into the integration of deep learning into the domain of AI. Following that, we elucidate the rationale behind harnessing deep learning for NLP tasks. In the domain of sentiment classification for drug reviews, the predominant approach has been the utilization of deep-learning algorithms.

This work investigates the effectiveness of the deep learning algorithms depending on the classification of sentiment analysis of drugs in the whole review text, conditions, and the rating score. It also focuses on comparing the effectiveness of bidirectional LSTM, and hybrid model (bidirectional LSTM-CNN). The investigation into the word embedding using GloVe focused on assessing its impact on model performance. Incorporating GloVe embeddings into the hybrid model holds the potential to significantly enhance performance by leveraging the semantic knowledge encoded in pre-trained word vectors. This integration augments the model's capacity to understand and represent the nuanced meanings of words within the context of the bidirectional LSTM-CNN architecture. By harnessing the rich semantic information captured by GloVe, the hybrid model becomes more adept at capturing both sequential dependencies and local patterns, ultimately contributing to improved overall performance. For evaluating and testing the performance the accuracy, precision, recall, and F1-score were applied to demonstrate the superiority of deep learning techniques in the task of sentiment analysis. Additionally, we used the testing data in two ways. For the drug review, we tested the drug reviews directly, and for WebMD, the data was split into a training set and a test set.

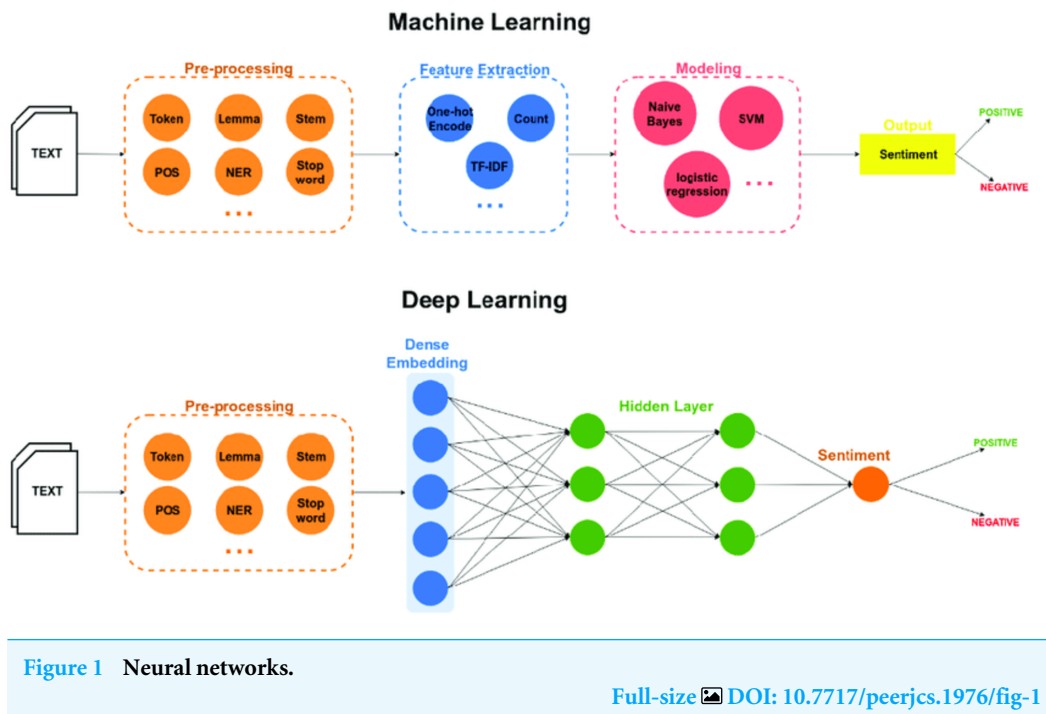

**Figure 1** Neural networks.

## Neural networks

Deep learning involves a multilayer approach to the hidden layers of neural networks (*Dang, Moreno-García & De la Prieta, 2021*). Traditionally, in machine learning models, features are identified and extracted either manually or using feature selection methods. However, in the case of deep learning (*Ghulam et al., 2019*), features are learned, and extracted automatically resulting in higher accuracy and performance. Figure 1 shows the differences in sentiment classification between the two approaches.

## Recurrent neural networks

A recurrent neural network (RNN) is a class of artificial neural networks where connections between nodes can create a cycle, allowing output from some nodes to affect subsequent input to the same nodes as shown in Fig. 2 This unique architecture allows RNNs to exhibit temporal dynamic behavior.

Specifically designed for processing sequential data, RNNs play a crucial role in various applications. The fundamental equation governing the recurrent nature of these networks is expressed as follows: RNN are a part of the neural network family used for processing sequential data. In the following equation

$$h^t = f(h^{t-1}, x), \tag{1}$$

$h^t$ represents the output at time $t$, $h^{(t-1)}$ denotes the previous output, and $x$ is the input. The function $f$ captures the recurrent relationship, describing how the network processes and updates information over sequential steps.

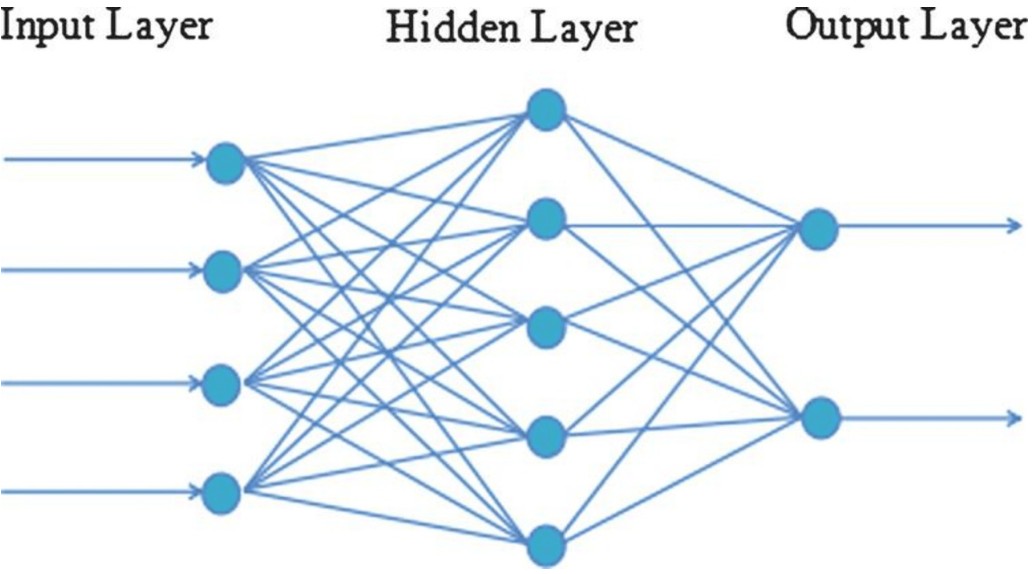

**Figure 2  RNN architecture.**

## Recurrent neural networks (long short-term memory)

The LSTM network is a type of RNN that includes additional memory features to address the challenges of vanishing and exploding gradients (*Ghulam et al., 2019*). LSTMs consist of recurrently connected blocks, known as memory units, that are designed to mitigate these issues. The memory units have gates, which determine whether to add or remove information from the cell state as shows in Fig. 3. By utilizing these gates, LSTMs can capture long-range dependencies in the input data, which is useful for text classification tasks. Overall, LSTMs offer an advanced approach to RNNs that can better handle complex and sequential data.

The architecture includes several gates and memory cells that enable it to capture and remember information over varying time scales (*Tran et al., 2016*). The LSTM has an input $x_t$ which can be the output of a CNN or the input sequence directly. $h_{t-1}$ and $c_{t-1}$ are the inputs from the previous timestep LSTM. $o_t$ is the output of the LSTM for this timestep. The LSTM also generates the $c_t$ and $h_t$ for the consumption of the next time step LSTM

- Forget gate: The forget gate determines what information from the previous cell state should be discarded (*Kim et al., 2016*). It takes the previous cell state $C_{t-1}$ and the current input $x_t$ as inputs and produces a value between 0 and 1 for each element in the cell state

$$f_t = \sigma(W_f \cdot [h_{t-1}, x_t] + b_f). \tag{2}$$

- Input gate: The input gate controls how much new information should be added to the cell state. It calculates two values: the candidate values $\tilde{C}_t$ (new candidate cell state) and the input gate $i_t$ which determines how much of this candidate value should be added.

$$i_t = \sigma(W_i \cdot [h_{t-1}, x_t] + b_i) \tag{3}$$

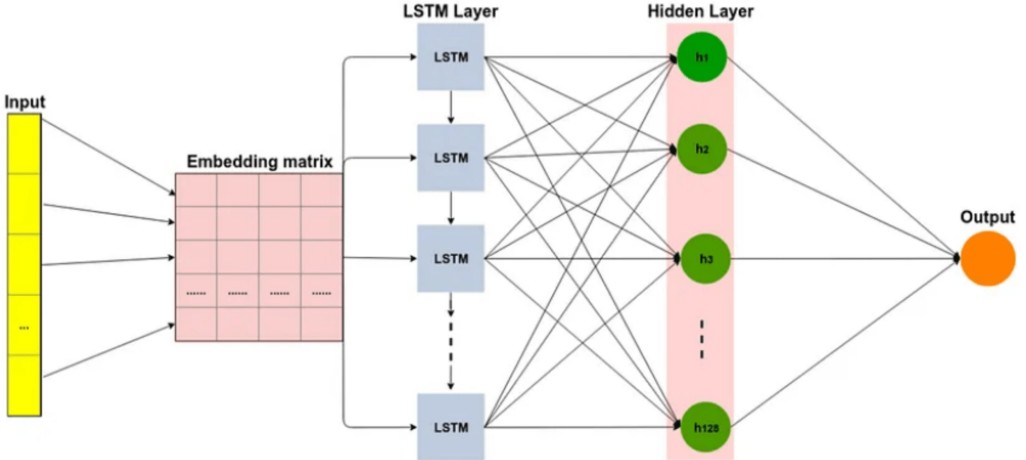

**Figure 3** **Long short-term memory (LSTM) architecture.**

$$\tilde{C}_t = \tanh(W_C \cdot [h_{t-1}, x_t] + b_C). \tag{4}$$

- Update cell state: The new cell state $C_t$ is calculated by combining the forget gate, the input gate, and the previous cell state.

$$C_t = f_t \cdot C_{t-1} + i_t \cdot \tilde{C}_t. \tag{5}$$

- Output gate: The output gate determines the next hidden state $h_t$ and the output of the LSTM cell. It combines the current input $x_t$ and the previous hidden state $h_{t-1}$ to calculate the output gate $o_t$ and the output

$$o_t = \sigma(W_o \cdot [h_{t-1}, x_t] + b_o) \tag{6}$$

$$h_t = o_t \cdot \tanh(C_t). \tag{7}$$

## Bidirectional LSTM

A bidirectional LSTM, or Bi-LSTM, is a sequence processing model that consists of two LSTMs as shows in Fig. 4: one taking the input in a forward direction, and the other a backward direction. Bi-LSTMs effectively increase the amount of information available to the network, improving the context available to the algorithm (*e.g.*, knowing what words immediately follow and precede a word in a sentence).

The architecture of bidirectional LSTM comprises of two unidirectional LSTMs which process the sequence in both forward and backward directions (*Graves & Schmidhuber, 2005*). This architecture can be interpreted as having two separate LSTM networks, one gets the sequence of tokens as it is while the other gets in the reverse order. Both of these LSTM networks return a probability vector as output, and the final output is derived from

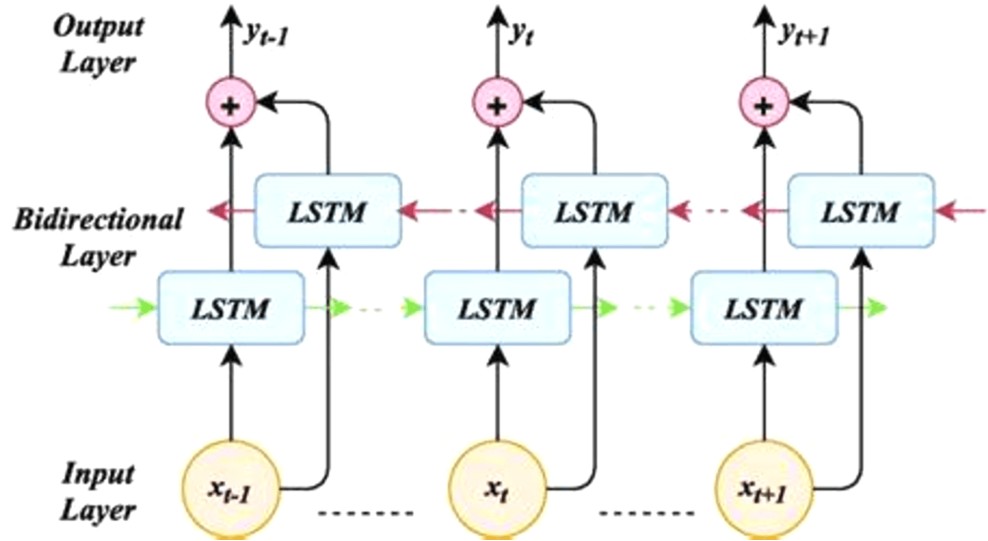

**Figure 4** Bidirectional LSTM (architecture).

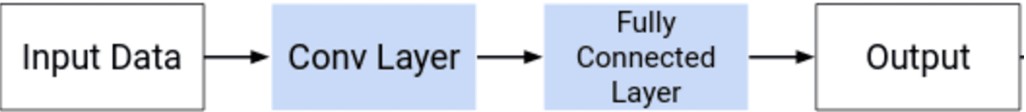

**Figure 5** Convolution neural networks.

the combination of these probability vectors. It can be represented as:

$$p_t-=p_t^f+p_t^b,\tag{8}$$

where

- $p_t$ represents the final probability vector of the network.
- $p_t^f$ represents the probability vector from the forward LSTM network.
- $p_t^b$ represents the probability vector from the backward LSTM network.

## Convolution neural networks

CNNs (*Moskovitz, Roy & Pillow, 2018*) are a type of deep neural network primarily used for analyzing visual imagery. Recently, text classification has been revolutionized by the advancements in deep learning particularly RNN and CNN. In fact, CNN for text classification is the new sexy for NLP practitioners in industry, research, and academia because of the efficient encoding of language semantics and context into mathematical vectors. In our model we proposed how the CNNs offer a powerful approach to analysis of visual data, leveraging their ability to learn and extract features from images. The

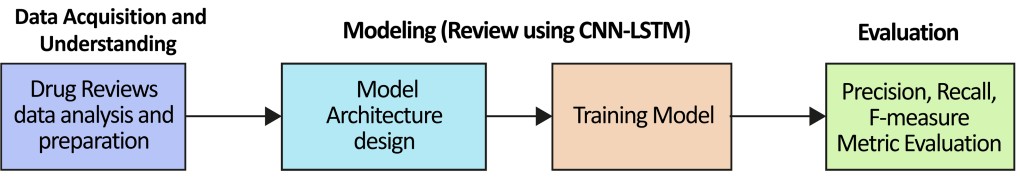

**Figure 6** TDSP level design flow prepared for this study.

architecture of a CNN can be deconstructed into two fundamental components as listed below and shows in Fig. 5:

- Convolutional layers: These layers are responsible for extracting features from the input data.
- Fully connected (dense) layers: After the convolutional layers identify relevant features, the fully connected layers utilize this extracted information to generate the final output.

## METHODS FOR SENTIMENT ANALYSIS OF PATIENTS' DRUG REVIEWS

This section outlines how various techniques and models are utilized to develop a robust and effective sentiment analysis system. First, the design flow of this research was descriptive and for each block in the flow, the diagram is explained in detail. The data sets used in this work, their source, and their components are then described. The relevant fields in the data set that were chosen for this research are described in detail. Data acquisition and understanding is the next phase and includes the pre-processing of the data and analysis. The data prepossessing techniques are also discussed. The next phase is modeling which is defined by the algorithms used, and it has three sub-stages: the CNN-LSTM model design for text quantification, model implementation, and finally, the evaluation model is described by precision, classifier recall, and F1-measure metric. These stages are shown in Fig. 6.

### Patients' drug reviews datasets

Two relevant datasets were planned to be used for this research. Both are publicly available drug reviews.

### *Drug*

The drug review dataset is shown in Table 1, and includes patient reviews on particular drugs, their associated medical conditions, and a 10-star satisfaction rating. It was utilized in the Winter 2018 Kaggle University Club Hackathon and is currently accessible to the public. This dataset comprises 51,408 instances and six attributes. Training and testing data were split as 75–25 in the present study. The number rating for the drug has been divided into three general classes: positive (7–10), negative (1–4), or neutral(4-7).

The UCI ML Drug Reviews dataset comprises seven features; however, for this study, particular emphasis is placed on three key features: condition, review, and rating. These specific features are deemed more crucial and impactful within the context of the study, as

**Table 1  Drug review dataset.**

| Field | Condition | Review | Rating |
|---|---|---|---|
| Definition | Health condition | User comment | User rating on 10-point scale |
| Type | Textual | Textual data | Numerical |
| | Drug review contains 161,297 reviews and seven attributes | | |

**Table 2  WebMD dataset.**

| Filed | Condition | review | Rating |
|---|---|---|---|
| Definition | Health condition | User comment | User rating on 5-point scale |
| Type | Textual | Textual data | Numerical |
| | WebMD continues 362,806 reviews and 12 attributes | | |

they play a significant role in shaping the analysis and outcomes. The condition provides insights into the health context, the review encapsulates the subjective user experience, and the rating quantifies the overall assessment. Together, condition, review, and rating form the focal elements that contribute substantially to the study's objectives and findings.

### WebMD

The second dataset is WebMD Drug shows in Table 2 which is used to compare the result depending on the balance of review in the dataset. The dataset provides user reviews on specific drugs along with related conditions, side effects, age, sex, and ratings reflecting overall patient satisfaction. There are around 0.36 million rows of unique reviews which are updated until Mar 2020. The WebMD dataset encompasses twelve features; however, within the scope of this study, special attention is directed towards three pivotal features: condition, review, and satisfaction. These specific features have been identified as significantly influential in the study. Delving deeply into these three columns is imperative, as they collectively contribute essential information. It is crucial to thoroughly examine and analyze conditions, reviews, and ratings, as they hold paramount importance in shaping the study's insights and outcomes.

## Data cleaning and prepossessing

These stages were carried out after gaining a thorough understanding of the behavior and features of the selected dataset. It was observed that the user must have to write the drug's name and intake quantity in the review which was not necessary for this research work. Therefore drugs' names and quantity measurements were dropped. In addition, the condition and review text emerged in one file. Algorithm 1 explains the next steps which are text cleaning and preparation modules and have to be utilized as listed below:

- Remove any leading or trailing spaces (*Sekihara et al., 2016*). Removing characters or white spaces from the beginning or end of a string is the basic step to preprocessing the text in NLP which is achieved by using the strip function.
- Convert all the review's text into lowercase using the lower() function.

- Substitute any occurrence of a new line character. To perform a global search and replace, use a regular expression with a space using the sub() function.
- Remove any punctuation and special characters from each word. If the input string consists of punctuation, then we have to make it punctuation-free. Let us say the input string is $Student@' then we have to remove $ and @, furthermore, we have to print the plain string 'Student' which is free from any punctuation. More details can be found at *GeeksforGeeks (2023)*.
- Remove English stop words (*Shilpa et al., 2021*), which are words that are very frequent in a language and are thus considered to not contain much information on the analyzed text such as ''the'', ''which'', ''where'', ''and'', ''is'', ''how'', and ''who'' from the text using the NLK.corpus and remove any word having less than three characters since such words do not provide much information.
- Lemmatize each token (*Ingason et al., 2008*): which entails reducing a word to its canonical or dictionary form. The root word is called a 'lemma'.The method entails assembling the inflected parts of a word in a way that can be recognized as a single element. The process is similar to stemming but the root words have meaning.
- Stemming is a rules-based approach that produces variants of a root/base word (*Lovins, 1968*). In simple words, it reduces a base word to its stem word. This heuristic process is the simpler of the two as the process involves indiscriminate cutting of the ends of the words. Stemming helps to shorten the look-up time and normalize the sentences for a better understanding.

## Word embedding by using GloVe encoding

In this work, contextual encoding which is already trained on large text corpus using semantic encoding is utilized. The encoding GloVe will be used in the LSTM bidirectional model and in the hybrid model LSTM-CNN shows in Algorithm 2 and Fig. 7. GloVe stands for global vectors for word representation. It is an unsupervised learning algorithm developed by Stanford University (*Pennington, Socher & Manning, 2014*) for generating word embedding by aggregating a global word-word co-occurrence matrix from a corpus. GloVe (*Naili, Chaibi & Ghezala, 2017*) is a widely used pre-training method in natural language processing (NLP) for generating word embedding, which are dense vectors that represent the semantic meaning of words. These embeddings are useful in various NLP tasks and can significantly improve the performance of deep learning models. By incorporating GloVe embedding as input in CNN, RNN, and LSTM models, the semantic meaning of words can be captured, leading to accurate predictions (*Mitchell et al., 2008*). Therefore, using GloVe embedding in pertaining deep learning models can be a powerful technique to improve the performance of NLP tasks. The GloVe model is trained with a massive corpus and uses it for the mathematical conversion of the input text.

## DEEP LEARNING MODELS FOR SENTIMENT ANALYSIS OF PATIENTS' DRUG REVIEWS

In this section, the modeling of Bi-LSTM and Bi-LSTM-CNN architectures is described depending on the classification of sentiment analysis of drug review text, conditions, and

---

**Algorithm 1** Clean Text Function

---

1: **function** CLEAN_TEXT(text)
2:                                                          ▷ Remove Pre and Post Spaces
3:     text ← strip(text)
4:                                                          ▷ Lower case the entire text
5:     text ← to_lower(text)
6:                                                          ▷ Substitute New Line Characters with spaces
7:     text ← substitute(text)
8:                                                          ▷ Tokenize the sentence
9:     word_tokens ← tokenize(text)
10:                                                         ▷ Initialize cleaned text list
11:     cleaned_text ← []
12:                                                         ▷ Remove punctuation and special characters
13:     **for** word **in** word_tokens **do**
14:         cleaned_word ← ""
15:         **for** char **in** word **do**
16:             **if** IS_ALPHANUMERIC(char) **then**
17:                 cleaned_word ← cleaned_word + char
18:             **end if**
19:         **end for**
20:         cleaned_text.append(cleaned_word)
21:     **end for**
22:                                                         ▷ Specify the stop words list
23:     stop_words ← get_stopwords($'english'$)
24:                                                         ▷ Initialize text_tokens list
25:     text_tokens ← []
26:                                                         ▷ Remove stopwords and words containing less than 2 characters
27:     **for** word **in** cleaned_text **do**
28:         **if** len(word) > 2 **and** word ∉ stop_words **then**
29:             text_tokens.append(word)
30:         **end if**
31:     **end for**
32:                                                         ▷ Lemmatize each word in the word list
33:     text ← lemmatize(text_tokens)
34:     **return** text
35: **end function**

---

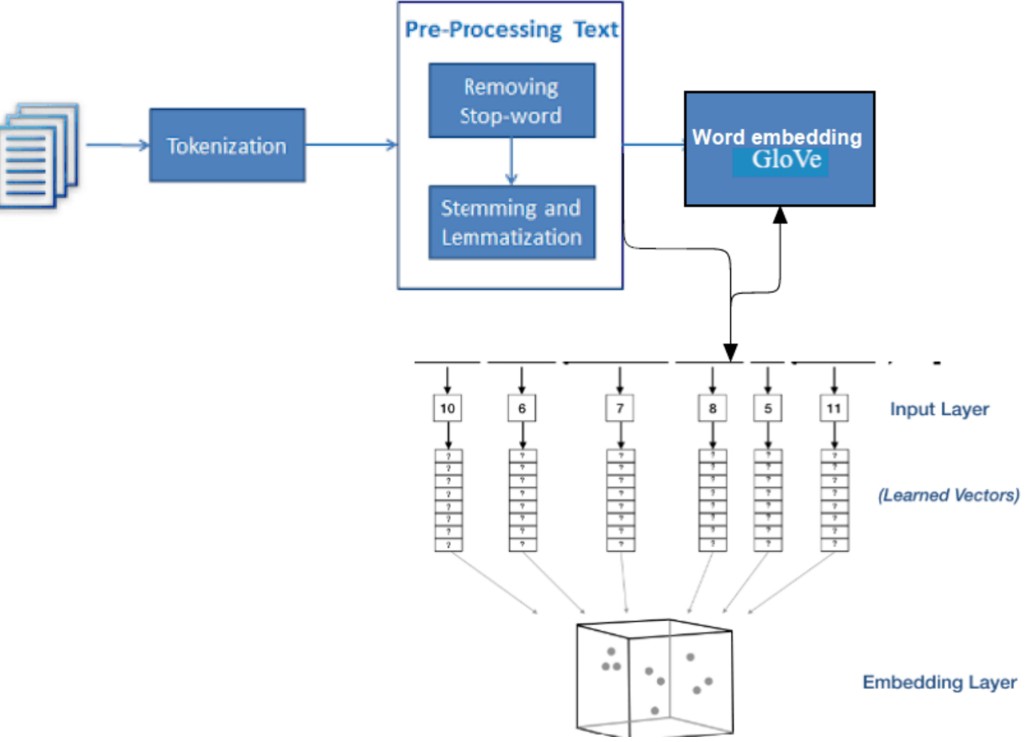

**Figure 7    Architectures for generating the embedding matrix.**

**Table 3    Modeling description.**

|  | Words embedding | Data training | Data testing |
|---|---|---|---|
| Model(A) | Without using GloVe encoding | Drugs train-raw | Drugs test-raw |
| Model(A) | Using GloVe encoding (GloVe.6B.300d) | Drugs train-raw | Drugs train-raw |
| Model(B) | Without using GloVe encoding (GloVe.6B.300d) | WebMD | WebMD |
| Model(B) | Using GloVe encoding (GloVe.6B.300d) | WebMD | WebMD |

the rating score of drug reviews. In line with the purpose of this work, the model is divided into two parts Model A which will be trained and tested in Drugs-Train, and Model B which will be trained and tested in WebMD as shown in Table 3.

Within this study, by integrating bidirectional LSTM and bidirectional LSTM-CNN models, along with GloVe embeddings, the research exemplifies the potential of deep learning-based approaches to propel the field of drug sentiment analysis within the healthcare domain. The emphasis is placed on utilizing these advanced techniques to conduct a detailed and nuanced analysis of sentiment in patients' drug reviews. This methodological approach aims to pave the way for advancements in understanding sentiment within the context of healthcare, setting a foundation for subsequent evaluation.

---

**Algorithm 2** Create Embedding Matrix Function

1: **function** CREATE_EMBEDDING_MATRIX($FP, WI, ED$)
2:                            ▷ File path, word index, embedding dimension
3:     $FP \leftarrow$ "GloVe.6B.300d"                              ▷ File path
4:     $WI \leftarrow word\_index$                                ▷ Word index
5:     $ED \leftarrow embedding\_dim$                         ▷ Embedding dimension
6:                                         ▷ Vocabulary size
7:     $vocab\_size \leftarrow \text{length}(word\_index) + 1$
8:                             ▷ Initialize the embedding matrix with zeros
9:     $embedding\_matrix \leftarrow \text{zeros}((vocab\_size, embedding\_dim))$
10:                                   ▷ Open the embedding file
11:     **with** open($file\_path$, encoding='utf-8') **as** file:
12:     **for** every line in file: **do**
13:                           ▷ Extract the word and its corresponding vector
14:         $word, *vector \leftarrow \text{line.split}()$
15:         **if** the word is in word_index: **then**
16:                                   ▷ Get the word index
17:             $IDX \leftarrow \text{word\_index}[word]$
18:                               ▷ Extract the embedding vector
19:             $embedding\_matrix[IDX] \leftarrow \text{array}(vector)[:embedding\_dim]$
20:         **end if**
21:     **end for**
22:     **return** $embedding\_matrix$
23: **end function**

---

## BiLSTM-CNN

Figure 8 shows the detailed flow diagram of all the steps within the modeling phase.

- The review underwent pre-training and embedding using GloVe vectorization to generate the embedding matrix, which was then fed into a dedicated convolution layer.
- Bi-LSTM: The output of the max pooling layer is then passed through a bidirectional LSTM layer. Hence, can be used to understand the long-distance relationship between the regions and predict the ratings effectively.
- A convolutional layer is added to extract features from the embedded sequences.
- Max pooling layer is added to reduce dimensionality.
- Finally, a dense layer with softmax activation is added to predict the sentiment class of the input sequence.

## Bidirectional LSTM model

As shown in Fig. 9 a bidirectional LSTM layer is added to capture the sequential dependencies in the data. Finally, the output layer is added with three units, one for each class in the classification task. The softmax activation function is used to output class probabilities.

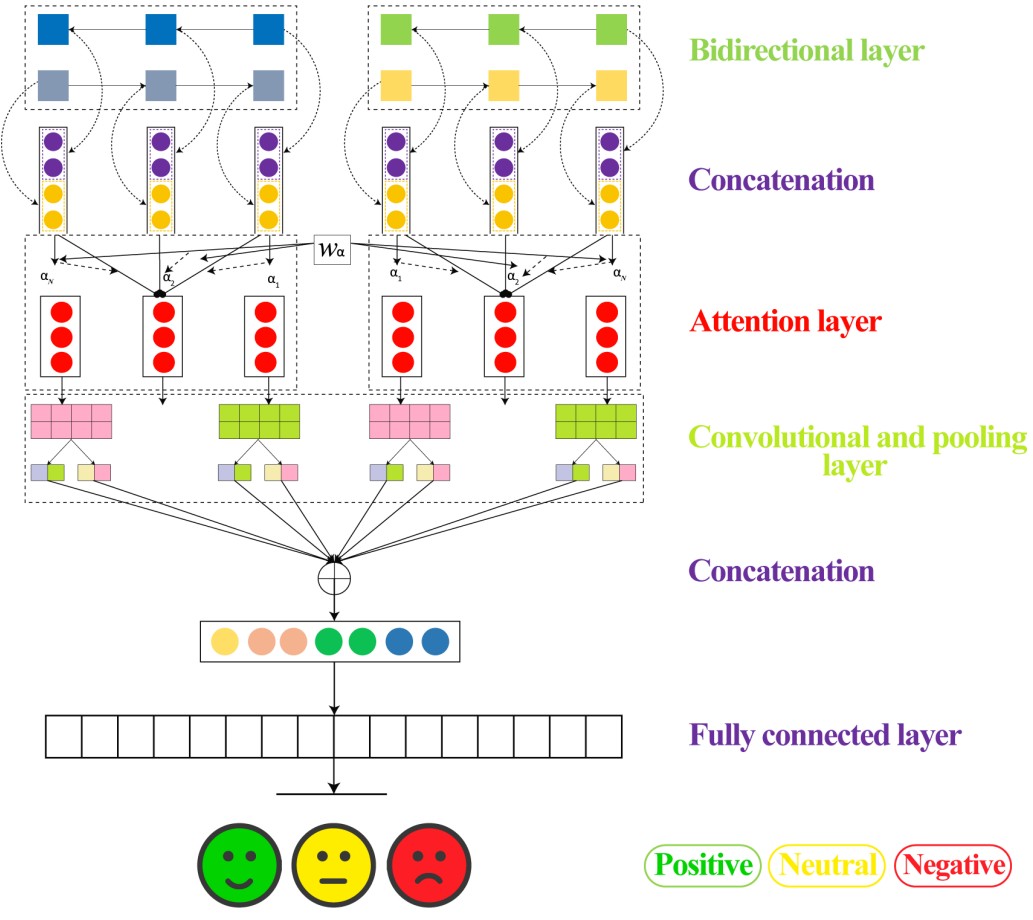

**Figure 8** Architectures for modeling LSTM-CNN.

## The architecture of the model

For both Model A and Model B, the Bi-LSTM and Bi-LSTM-CNN architectures are applied as explained below:

- The model starts with the embedding layer which creates a dense vector representation of each word in the input sequence. It has three arguments the size of the vocabulary, output dimensions, and the length of the input sequence.
- The second layer is the SpatialDropout1D layer which helps prevent over-fitting.
- A bidirectional LSTM layer was applied enabling the input sequence to be applied in both forward and backward directions. This bidirectional processing allows the model to glean insights from the entire context, capturing dependencies not only from preceding words but also from subsequent ones. Consequently, the bidirectional LSTM layer enables the model to discern complex patterns and nuances in drug reviews, facilitating a more nuanced sentiment analysis. The inclusion of a SpatialDropout1D layer is crucial for preventing overfitting, enhancing the model's generalization capabilities by selectively dropping entire 1D feature maps during training

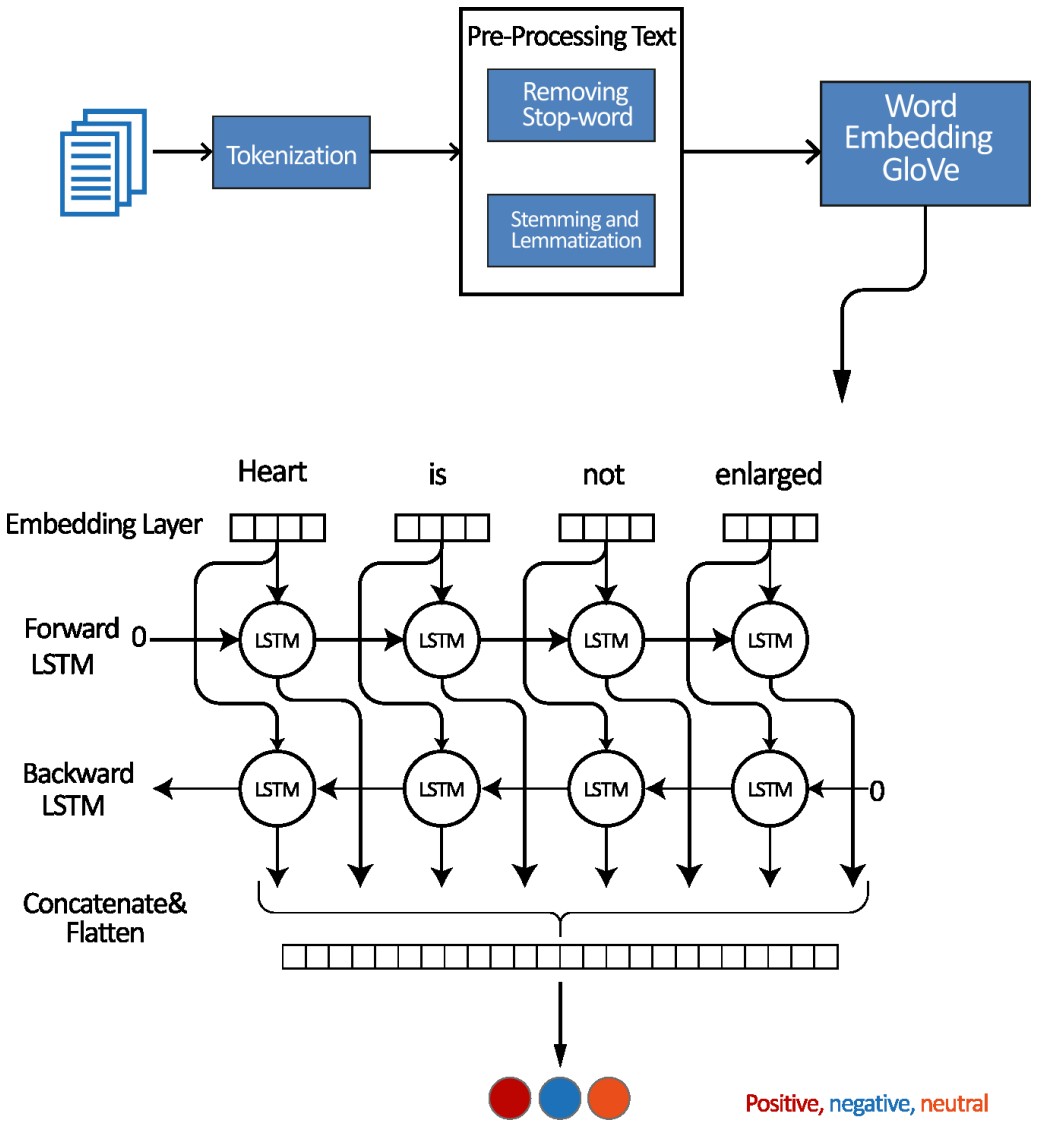

**Figure 9** Architectures for modeling LSTM-Bil.

- The dense layer applies a linear transformation to the output of the previous layer, followed by a softmax activation function. The softmax function normalizes the output to a probability distribution over the three classes. The concluding Dense layer, coupled with a softmax activation function, serves as the ultimate decision-making layer. It transforms the output from preceding layers into a probability distribution across the three sentiment classes
- The method configures the learning process for the model by compile processing which was used in programming contexts. The loss argument specifies the loss function to optimize, the optimizer argument specifies the optimization algorithm and the metrics argument specifies the evaluation metric(s) to use during training and testing.

The bidirectional LSTM layer in the Bi-LSTM-CNN architecture maintains its role in considering both past and future context, synergizing with the convolutional layer's local feature extraction. This combination proves particularly effective in capturing intricate nuances in the sentiment expressed within drug reviews. Similar to the Bi-LSTM architecture, the Dense layer, and compile processing are integral components for transforming learned features into actionable predictions, ensuring optimal learning and evaluation processes. In summary, the Bi-LSTM-CNN architecture amalgamates the strengths of both LSTM and CNN components, providing a holistic approach to sentiment analysis in drug reviews.

### Architecture A and B Bi-LSTM-CNN

The architecture of Models A and B Bi-LSTM-CNN are the followings:

- The model starts with the embedding layer which creates a dense vector representation of each word in the input sequence. it has three arguments the size of the vocabulary, output dimensions, and the length of the input sequence).
- In the second reiteration, the arguments for the Embedding layer will be four because adding the weights parameter specifies the pre-trained word embedding by using GloVe encoding GloVe.6B.300d.
- The second layer is added as a 1D convolutional layer with 64 filters and a filter size of 5. The activation function used is ReLU (rectified linear unit).
- Max pooling layer with a pool size of 4 was applied to reduce the dimensionality of the data.
- A bidirectional LSTM layer was applied with 64 units, a dropout rate of 0.1, and a recurrent dropout rate of 0.3. The bidirectional LSTM allows the model to consider both past and future context in the input data.
- The dense layer applies a linear transformation to the output of the previous layer, followed by a softmax activation function. The softmax function normalizes the output to a probability distribution over the three classes
- The method configures the learning process for the model by compile processing. The loss argument specifies the loss function to optimize, the optimizer argument specifies the optimization algorithm and the metrics argument specifies the evaluation metric(s) to use during training and testing.

## RESULTS

The results in Table 4 show that the hybrid model LSTM-CNN can achieve a higher accuracy. Model A is applied to the Drug Review dataset, and Model B is applied to the Dataset in WebMD Drug. The results for accuracy show that using global vectors for word representation (GloVe) generally improved the performance of Model B and had the opposite effect in Model A. Model A (Bi-LSTM) has an accuracy of 85% without GloVe embedding and only 76% with GloVe embedding. While, Model B (Bi-LSTM) has an accuracy of 74% without GloVe embeddings and 75% with GloVe embedding.

Model A (Bi-LSTM-CNN) has an accuracy of 96% without GloVe embeddings and 87% with GloVe embedding. As well, Model B (BiLSTM-CNN) has an accuracy of 87% with

**Table 4  Evaluation accuracy with word embeddings with glove and without it for drug review dataset for both Model A-B.**

|  | Model A | | Model B | |
|  | Accuracy | | Accuracy | |
|  | Without GloVe | With GloVe | Without GloVe | With GloVe |
| --- | --- | --- | --- | --- |
| Bi-LSTM | 0.85 | 0.76 | 0.74 | 0.75 |
| LSTM-CNN | 0.96 | 91 | 0.87 | 0.88 |

**Table 5  Bi-LSTM without GloVe embedding.**

|  | Model A without GloVe | | | Model B without GloVe | | |
| – | Precision | Recall | F1-score | Precision | Recall | F1-score |
| --- | --- | --- | --- | --- | --- | --- |
| Negative | 0.78 | 0.84 | 0.81 | 0.76 | 0.80 | 0.78 |
| Neutral | 0.00 | 0.00 | 0.00 | 0.68 | 0.04 | 0.08 |
| Positive | 0.87 | 0.96 | 0.92 | 0.73 | 0.91 | 0.81 |
| Accuracy |  | 0.85 |  | 0.74 |  |  |
| Macro avg | 0.55 | 0.60 | 0.57 | 0.72 | 0.58 | 0.56 |
| Weighted avg | 0.77 | 0.85 | 0.81 | 0.74 | 0.74 | 0.70 |
| Accuracy |  | 0.847564 |  |  | 0.744680 |  |
| Cohen-score |  | 0.660347 |  |  | 0.546443 |  |

GloVe embeddings and 87% without GloVe embedding. Although the GloVe can improve the performance of the model, in our investigation, the GloVe achieved poor improvement in the performance Model A. The main reason behind this is we have enough records here (100,000 to be exact) for our embedding layer in the Encoder-Decoder model to learn the semantics of the language, so it performed quite well even without the pre-trained embeddings.

## Bi-LSTM without GloVe embedding

Based on the evaluation metrics provided in Table 5 above, and Figs. 10 and 11 parts a and b, respectively, it seems that Model A without GloVe outperforms Model B without GloVe in terms of precision, recall, F1-score, and accuracy for all three classes. The precision, recall, and F1-score for the neutral class are 0 for Model A without GloVe, indicating that the model does not predict this class well. However, Model B without GloVe has a higher precision, recall, and F1-score for the Neutral class, which means it can predict this class better than Model A without GloVe. The overall accuracy for Model A without GloVe is 0.847564, which is higher than the accuracy of Model B without GloVe at 0.744680. Additionally, the Cohen-score for Model A without GloVe is 0.660347, which is higher than the Cohen-score for Model B without GloVe at 0.546443. Therefore, we can conclude that Model A without GloVe performs better than Model B without GloVe in terms of overall sentiment classification for drug reviews.

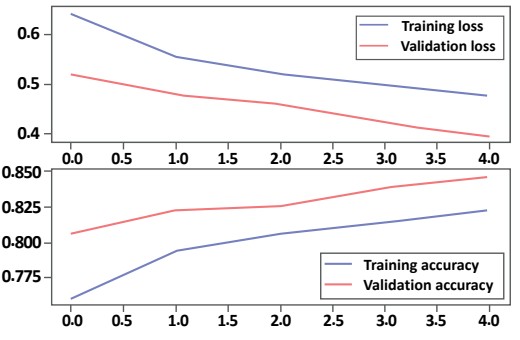
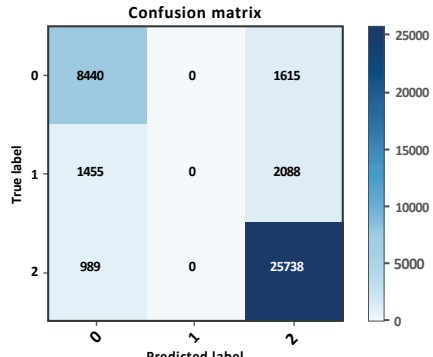

**Figure 10** LSTM loss and accuracy curves and confusion matrix Model A without GloVe embedding.

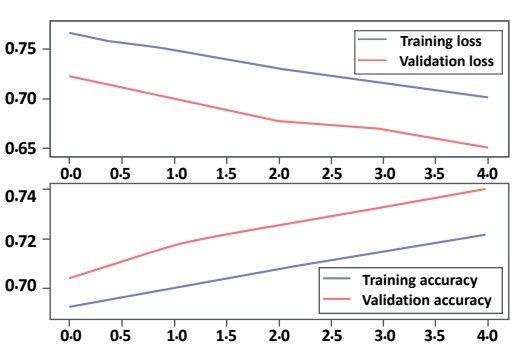
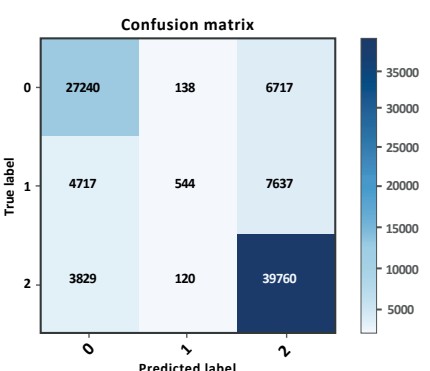

**Figure 11** LSTM loss and accuracy curves and confusion matrix Model B without GloVe embedding.

### Bi-LSTM-CNN without GloVe embedding

Overall, Model A outperformed Model B on all evaluation metrics, indicating that Model A is a better-performing model. As show in Table 6 and Figs. 12, 13 parts a and b, respectively. Model A performed well in all classes, achieving high precision, recall, and F1-scores on all sentiment classes, indicating that the model is performing well. The accuracy score for Model A is also higher, indicating that the model is performing better overall. Cohen's kappa scores for both models are relatively high, indicating that the models have a high level of agreement with the actual sentiment labels. Overall, depending on the result, the model Bi-LSTM-CNN achieved height accuracy on both models with GloVe embedding and without embedding than Bi-LSTM, and the Cohen score is perfect.

### Bi-LSTM with GloVe embedding

From the evaluation results in Table 7 above and Figs. 14, 15 parts a and b, respectively, it can be seen that the models with GloVe embeddings (Model B) outperformed the models without GloVe embeddings (Model A) in terms of precision, recall, and F1-score. However, there is a trade-off between precision and recall for the negative class, as the models with GloVe embeddings had higher precision but lower recall compared to the models without

**Table 6  Evaluation of Bi-LSTM-CNN without GloVe embedding.**

| – | Model A without GloVe | | | Model B without GloVe | | |
|---|---|---|---|---|---|---|
| | Precision | Recall | F1-score | Precision | Recall | F1-score |
| Negative | 0.98 | 0.97 | 0.97 | 0.90 | 0.89 | 0.90 |
| Neutral | 0.92 | 0.90 | 0.91 | 0.88 | 0.90 | 0.89 |
| Positive | 0.99 | 0.99 | 0.87 | 0.88 | 0.90 | 0.89 |
| Accuracy | | 0.98 | | 0.91 | | |
| Macro avg | 0.96 | 0.96 | 0.96 | 0.86 | 0.80 | 0.82 |
| Weighted avg | 0.98 | 0.98 | 0.98 | 0.87 | 0.87 | 0.87 |
| Accuracy | | 0.971581 | | | 0.874666 | |
| Cohen-score | | 0.941360 | | | 0.958807 | |

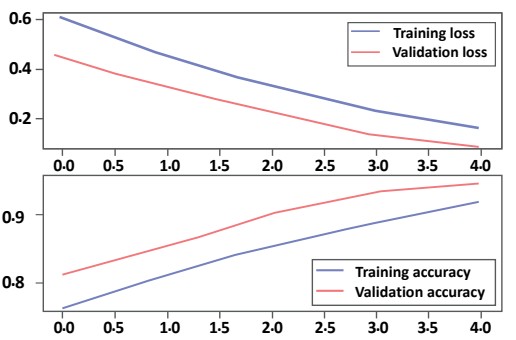 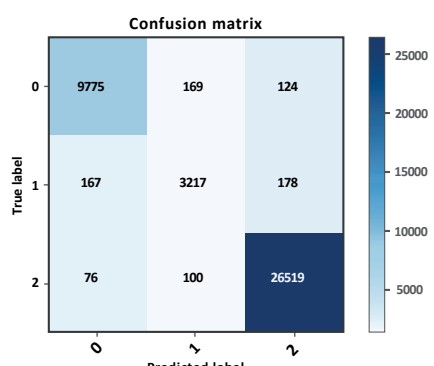

**Figure 12  Bi-LSTM-CNN loss and accuracy curves and confusion matrix Model A without GloVe.**

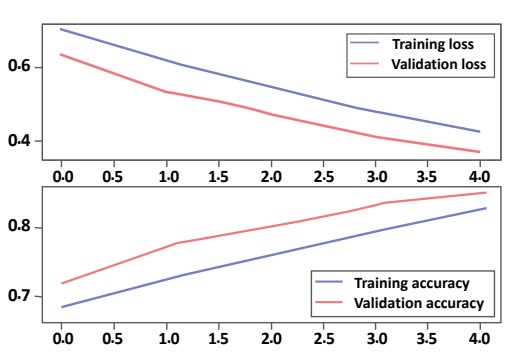 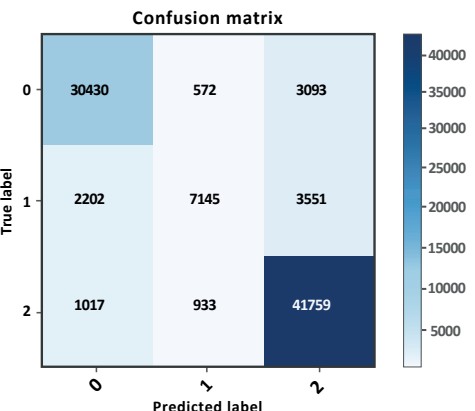

**Figure 13  Bi-LSTM-CNN loss and accuracy curves and confusion matrix Model B without GloVe embedding.**

**Table 7** Bi-LSTM with GloVe embedding.

| – | Model A with GloVe | | | Model B with GloVe | | |
|---|---|---|---|---|---|---|
| | Precision | Recall | F1-score | Precision | Recall | F1-score |
| Negative | 0.81 | 0.42 | 0.56 | 0.77 | 0.81 | 0.79 |
| Neutral | 0.00 | 0.00 | 0.00 | 0.66 | 0.06 | 0.11 |
| Positive | 0.75 | 0.98 | 0.85 | 0.74 | 0.91 | 0.82 |
| Accuracy | | 0.85 | | 0.75 | | |
| Macro avg | 0.52 | 0.47 | 0.47 | 0.72 | 0.59 | 0.57 |
| Weighted avg | 0.70 | 0.76 | 0.71 | 0.74 | 0.75 | 0.70 |
| Accuracy | | 0.760198 | | | 0.750711 | |
| Cohen-score | | 0.378105 | | | 0.558728 | |

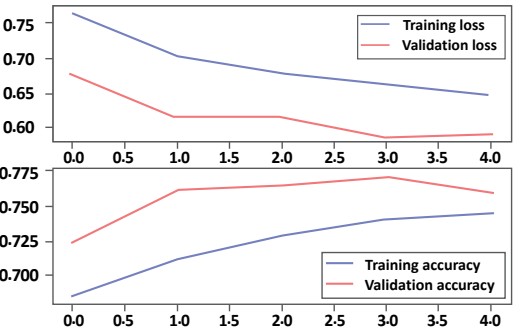
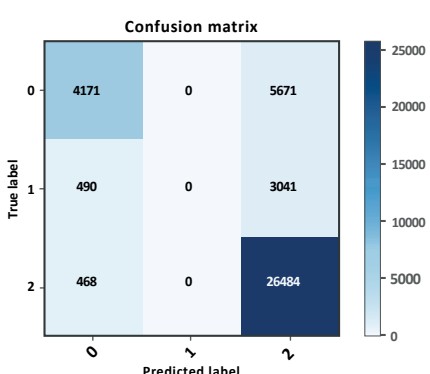

**Figure 14** LSTM loss and accuracy curves and confusion matrix Model A with GloVe embedding.

GloVe embeddings. The neutral class had poor performance overall, with F1-scores close to zero, while the positive class had the best performance. The accuracy of the models was similar both with and without GloVe embeddings. The Cohen score was higher for Model B, which indicates better inter-rater agreement. Overall, the results suggest that using GloVe embeddings can improve the performance of sentiment classification models on drug reviews from patients.

## Bi-LSTM-CNN with GloVe embedding

Based on the evaluation metrics in Table 8 and Figs. 16 and 17 parts a and b respectively, it can see the LSTM-CNN algorithm with GloVe embeddings performed better on Model A than on Model B. The LSTM-CNN algorithm with GloVe embeddings performed well on both models, with higher scores on Model A compared to Model B. The model achieved high precision, recall, and F1 scores on all sentiment classes in Model A, indicating that the model is performing well in all classes. However, the performance Model B was lower for the neutral sentiment class, indicating that the model is having some difficulty correctly classifying neutral sentiment. The accuracy scores for both models are relatively high,

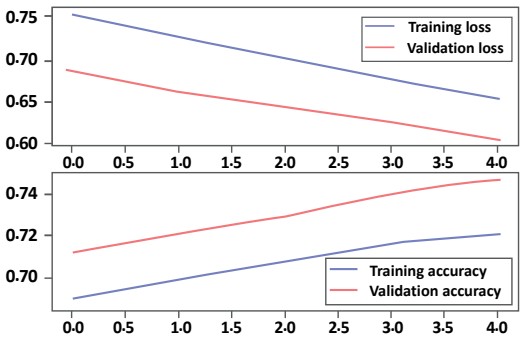
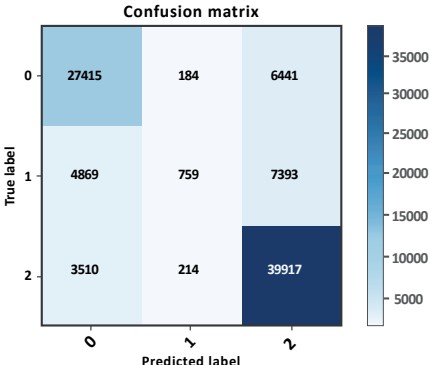

**Figure 15** LSTM loss and accuracy curves and confusion matrix Model B with GloVe embedding.

**Table 8** Evaluation of Bi-LSTM-CNN with GloVe embedding.

|  | Model A with GloVe | | | Model B with GloVe | | |
| --- | --- | --- | --- | --- | --- | --- |
| – | Precision | Recall | F1-score | Precision | Recall | F1-score |
| Negative | 0.86 | 0.93 | 0.89 | 0.93 | 0.88 | 0.90 |
| Neutral | 0.65 | 0.59 | 0.62 | 0.79 | 0.61 | 0.69 |
| Positive | 0.97 | 0.95 | 0.96 | 0.86 | 0.95 | 0.90 |
| Accuracy |  | 0.91 |  |  | 0.88 |  |
| Macro avg | 0.83 | 0.82 | 0.82 | 0.86 | 0.82 | 0.83 |
| Weighted avg | 0.91 | 0.91 | 0.91 | 0.88 | 0.88 | 0.87 |
| Accuracy |  | 0.913528 |  |  | 0.876342 |  |
| Cohen-score |  | 0.825912 |  |  | 0.791976 |  |

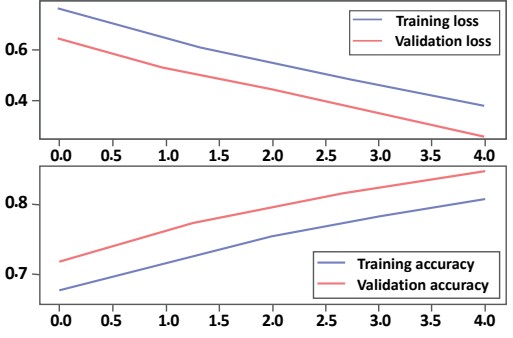
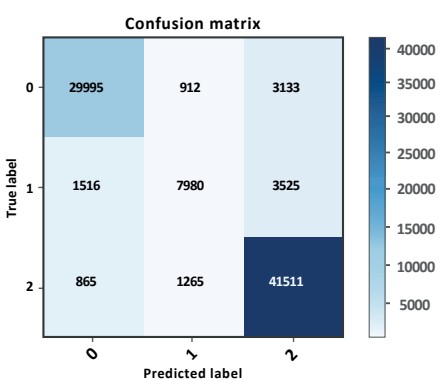

**Figure 16** Bi-LSTM-CNN loss and accuracy curves and confusion matrix Model A with GloVe embedding.

indicating that the model is performing well overall. The Cohen score was higher for Model B, which indicates better inter-rater agreement.

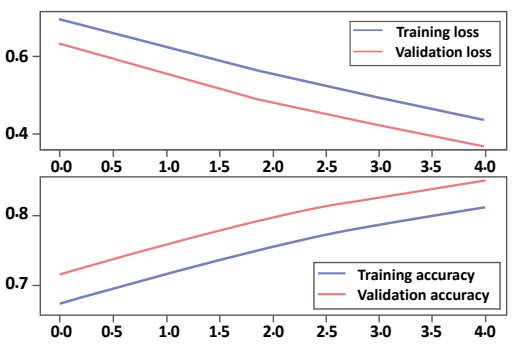 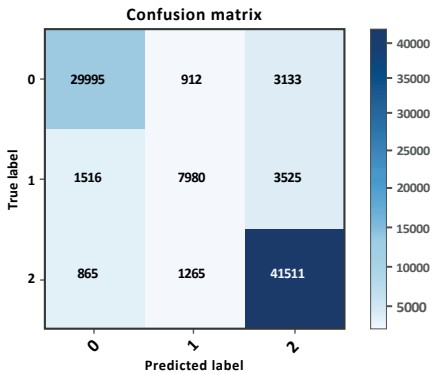

**Figure 17** Bi-LSTM-CNN loss and accuracy curves and confusion matrix Model B with GloVe embedding.

**Table 9** Cohen's Kappa interrater measurement.

| Kappa Value interpretation | |
|---|---|
| < 0 | No agreement |
| 0–.20 | Slight |
| .21–.60 | Fair |
| .41–.60 | Moderate |
| .61–.80 | Substantial |
| .81–1.0 | Perfect |

## COHEN'S KAPPA INTERRATER

Kappa measurement is an interrater reliability measure used to understand the agreement level between two rating sources and to establish whether any of those sources biased the results as shown in Table 9.

The Cohen's Kappa values were mainly substantial and perfect as shown in Tables 8 and 6.

## COMPARATIVE ANALYSIS

As Table 10 shows, in a previous study, *Yadav & Vishwakarma (2020)* delved into a text-representation framework grounded in weighted word embeddings, utilizing a combination of TF-IDT weighing scheme and FastText word embeddings. Notably, the SVM classifier emerged as the most effective, achieving a remarkable F1-score of 91.7%. Another significant study by *Thoomkuzhy (2020)* examined the comparative efficacy of the regional CNN-LSTM architecture against in-domain approaches. While regional CNN-LSTM demonstrated superior performance, the study fell short of attaining its predetermined research objectives. *Gräßer et al. (2018)* contributed to the discourse by employing SVM classification, resulting in a Cohen score of 83%. Their research underscored the inherent challenge of surpassing an 82% Cohen score, a metric measuring interrater agreement between given and computer-predicted ratings for a set of drugs.

**Table 10  Comparative analysis on sentiment analysis of patients' drug reviews.**

| Research | Dataset source | Model used | Predicted output | Accuracy |
|---|---|---|---|---|
| *Na & Kyaing (2015)* | WebMD.com | Clause-wise Lexicon approach and Classification using SVM | Sentiment polarity | 62.0 |
| *Gräßer et al. (2018)* | Drugs.com, Drugslib | Logistic regression | Drug rating | 92.24 |
| *Thoomkuzhy (2020)* | Drugs.com, Drugslib | Regression using regional CNN-LSTM | Drug rating/Sentiment Dimensionality | 65.07 |
| *Yadav & Vishwakarma (2020)* | Drugs.com | Classification using SVM, NB, DT, RF, KNN | Drug rating | 94.6 |
| This work | Drug.com, WebMD.com | Classification using In-Domain CNN-BILSTM | Drug rating | 97.1 |

Building upon these insights, this current research investigates the application of CNN-BiLSTM in the In-domain context. Our proposed model achieved a notable accuracy of 97%, F1 scores of 97% for negative, 91% for natural and 87% for positive, and a Cohen score of 94%, surpassing previous benchmarks. This study contributes valuable perspectives to the academic discourse on effective methodologies, particularly in employing CNN-LSTM, for sentiment analysis in patients' drug reviews. The findings presented herein offer valuable contributions to the ongoing exploration of advanced techniques in sentiment analysis within the healthcare domain.

# CONCLUSION

The experimental results showed that the Bi-LSTM-CNN model achieved the highest accuracy and Cohen score for sentiment analysis of patients' drug reviews in both cases: with GloVe embedding or without GloVe embedding. The Bi-LSTM-CNN model combines the strengths of both LSTM and CNN layers, allowing it to capture both temporal and spatial dependencies in the data. The CNN layers can extract local features from the input sequence, while the Bi-LSTM layers can capture longer-term dependencies and contextual information. This additional capacity has enabled the Bi-LSTM-CNN to learn more complex patterns in the data and make better predictions, leading to higher accuracy. Moreover, it can be concluded that using GloVe embedding does not show significant improvement in the performance of sentiment classification models. Therefore, the word embedding will be more useful if the regional LSTM-CNN model is used. Involving the regional CNN and LSTM models can be used for future work to extract drug rating or valence-arousal pairs by mining textual review data. To extend the analysis further, the implementation of drug reviews across many diseases snd conditions is required. Several considerations are worth reporting for future work such as the limited size of the available datasets. In addition, using the domain-specific vectorization of text data may help to improve the efficiency of the model.

### Funding

This work is funded by the Deanship of Scientific Research at Imam Mohammad Ibn Saud Islamic University (IMSIU) through Research Partnership Program no RP-21-07-09. The funders had no role in study design, data collection and analysis, decision to publish, or preparation of the manuscript.

### Grant Disclosures

The following grant information was disclosed by the authors:
The Deanship of Scientific Research at Imam Mohammad Ibn Saud Islamic University (IMSIU): RP-21-07-09.

### Competing Interests

The authors declare there are no competing interests.

### Author Contributions

- Sena Al-Hadhrami conceived and designed the experiments, performed the experiments, analyzed the data, performed the computation work, prepared figures and/or tables, authored or reviewed drafts of the article, and approved the final draft.
- Tamas Vinko performed the experiments, analyzed the data, performed the computation work, prepared figures and/or tables, authored or reviewed drafts of the article, and approved the final draft.
- Tawfik Al-Hadhrami conceived and designed the experiments, analyzed the data, performed the computation work, prepared figures and/or tables, authored or reviewed drafts of the article, and approved the final draft.
- Faisal Saeed conceived and designed the experiments, analyzed the data, performed the computation work, prepared figures and/or tables, authored or reviewed drafts of the article, and approved the final draft.
- Sultan Noman Qasem performed the experiments, analyzed the data, performed the computation work, prepared figures and/or tables, authored or reviewed drafts of the article, and approved the final draft.

### Data Availability

The code for both datasets is available at GitHub and Zenodo:

- https://github.com/senaalhadhrami/Deep-Learning-Based-Method-for_Patients-Drug-Reviews.git

- senaalhadhrami. (2024). senaalhadhrami/Deep-Learning-Based-Method-for_Patients-Drug-Reviews: v1.0.0 (v1.0.0). Zenodo. https://doi.org/10.5281/zenodo.10824129

The UCI ML Drug Review dataset is available at GitHub and Zenodo:

- https://github.com/senaalhadhrami/Drug_Review_dataset.git

- senaalhadhrami. (2024). senaalhadhrami/Drug_Review_dataset: Druge review dataset (v1.0.0). Zenodo. https://doi.org/10.5281/zenodo.10824426

The WebMD Dataset is available at GitHub and Zenodo:

- https://github.com/senaalhadhrami/Webmdataset1.git

- senaalhadhrami. (2024). senaalhadhrami/Webmdataset1: v1.0.0 (v1.0.0). Zenodo. https://doi.org/10.5281/zenodo.10824346

The GloVe embedding file is available at GitHub and Zenodo:

- https://github.com/senaalhadhrami/GloVe.git

- senaalhadhrami. (2024). senaalhadhrami/GloVe: GloVe word embeddings (v1.0.0). Zenodo. https://doi.org/10.5281/zenodo.10824438

## Supplemental Information

Supplemental information for this article can be found online at http://dx.doi.org/10.7717/peerj-cs.1976#supplemental-information.

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
