# Peer review of "Deep learning-based method for sentiment analysis for patients’ drug reviews"

_PeerJ Computer Science, doi:10.7717/peerj-cs.1976_

## Round 0.1 · original submission · Major Revisions

Based on the reviewers’ comments, you may resubmit the revised manuscript for further consideration. Please consider the reviewers’ comments carefully and submit a list of responses to the comments along with the revised manuscript.

**Language Note:** The review process has identified that the English language must be improved. PeerJ can provide language editing services - please contact us at copyediting@peerj.com for pricing (be sure to provide your manuscript number and title). Alternatively, you should make your own arrangements to improve the language quality and provide details in your response letter. – PeerJ Staff

Reviewer 1 ·

Basic reporting

The paper is well-written and aims to evaluate the effectiveness of bidirectional Long-term memory (LSTM) and a hybrid model (bidirectional LSTM-CNN) for sentiment classification of drug reviews. The study also investigated the impact of using GloVe word embeddings on the model’s performance. However, the reviewer's feedback should be addressed. Also, the below minor corrections should be considered:

- English typos should be corrected. Example: "Firstly, Investigates the effectiveness ..." , "Data Set ...", "Table 3 explains the next steps, which are text cleaning and preparation modules and must be utilized as listed below:"

- The citations should be checked and revised. Example: Kyaing et al. Na et al.(2012). line 79.

- Table 3. The Pseudo-code generation for the function clean text. This is not a table. Reformat it as an algorithm.

- Re-number all sections and its subsections. Example of the wrong numbering:
1 RELATED WORK
2 WORD EMBEDDING BY USING GLOVE ENCODING

Experimental design

The authors conducted several experiments and applied Convolution Neural Networks well. However, a few corrections should be made to improve the presentation of the Experimental Design.

- In Figure 5. Convolution Neural Networks, why the final output is the error? Please check and review this Figure.
- Why Table 4 has two columns? What are these columns? I suggest to reformat Table 4 as an algorithm.

Validity of the findings

Validation

The authors evaluated the proposed models using benchmark evaluation metrics such as accuracy, precision, recall, and F1-score. However, a few corrections are required:

1. Highlight (bold) the best values in each table.
2. The last sections should be re-ordered as follows:

Model without GloVe
Model with GloVe

3. All figures have the same titles. Please correct it.
Figure 12. LSTM loss and accuracy curves and Confusion matrix model A
Figure 13. LSTM loss and accuracy curves and Confusion matrix model B
Figure 14. LSTM loss and accuracy curves and Confusion matrix model A
Figure 15. LSTM loss and accuracy curves and Confusion matrix model B

4. The figures in (3.5 Bi-LSTM with Glove Embedding) and (3.6 Bi-LSTM-CNN with Glove Embedding) wrongly included :

LSTM without GloVe

Additional comments

Please review and correct all Figures' titles.

Cite this review as

Reviewer 2 ·

Basic reporting

This paper employs deep learning algorithms to conduct sentiment analysis on drug reviews. However, it notably omits a thorough exploration of existing research on drug sentiment analysis, only providing a broad overview of sentiment analysis studies in the literature survey. Recent studies utilizing the datasets presented in this research have been overlooked, leaving me perplexed as to why the authors chose not to delve into this crucial aspect.

Experimental design

The paper does not state how their study contributes in comparison to the other studies for drug sentiment analysis.

Validity of the findings

The paper lacks a comparative analysis between machine learning algorithms and deep learning algorithms to assess accuracy. The absence of such a comparison raises questions about the comprehensiveness of the study, as it overlooks an essential element in evaluating the performance of the employed algorithms.

Cite this review as

Reviewer 3 ·

Basic reporting

The deep learning-based method used in this paper has explored and reviewed the patient's drug reviews, the proposed methodology has involved the implementation and evaluation of bidirectional LSTM and bidirectional LSTM-CNN architectures for sentiment analysis of drug reviews. Contextual encoding, specifically the GloVe word embeddings, is leveraged for both models to enhance performance.
Following are some of the concerns the author should address:
1. English must be proofread
2. The experiments must be justified and wider details are required to support the claim.
3. The abstract should be focused, shortening the abstract is recommended
4. Recent papers must be cited in the introduction
5. The problem statement at the end of the related work section should be improved and expose the paper's idea
6. Numbering seems to have an issue with this paper.
7. Figures are not readable for example Figure 1.
8. Meaning titles should be assigned for example Method section the title should be Method of ....... related to the paper, as well modeling section
9. Figures and tables must be cited in the text and added after the citing in the first space and acceptable place.
10. Table 3 for Pseudo code must be enhanced and proper pseudo code must be added
11. More detail about the modeling scenarios in the results and discussion section, with justifications for the achieved results, why these results are linked to the current implementation, and the method that has been used in this study.
12. The conclusion is a bit long and it should be focused and short.

Experimental design

13. Give a brief description of the features of the two datasets that were used (drug reviews and WebMD). What is the size of the datasets, and are there any particular characteristics that might affect how well your models perform?

14. Describe the rationale for the selection of bidirectional LSTM and bidirectional LSTM-CNN for sentiment analysis in relation to medication reviews.
What benefits do these models have over alternative possible architectures?
15. it’s excellent that you mentioned using F1 score, accuracy, precision, and memory.
However, to provide a more thorough picture of model performance, think of reporting these metrics for each class (positive, negative, and neutral).

16. Provide a succinct analysis of the results. What does sentiment analysis for medicine reviews signify in relation to these accuracy numbers? Were there any particular issues or trends found in the reviews that affected the models' functionality?

Validity of the findings

Please thoroughly revise the language of the paper, for example, use "achieved an accuracy of 96%" instead of "achieves an accuracy of 96%" for past study results

Cite this review as

---

## Round 0.2 · accepted · Accept

Congratulations, the reviewers are satisfied with the revised version of the manuscript and have recommended the acceptance decision.

Reviewer 1 ·

Basic reporting

The authors incorporate all suggestions.

Experimental design

The authors incorporate all suggestions.

Validity of the findings

The authors incorporate all suggestions.

Additional comments

No comments

Cite this review as

Reviewer 3 ·

Basic reporting

Figures are improved according to the guidelines.

Experimental design

Experiments and methods discussion is updated in the relevant sections.

Validity of the findings

no comment

Cite this review as